# The Therapeutic Potential of Vitamins B1, B3 and B6 in Charcot–Marie–Tooth Disease with the Compromised Status of Vitamin-Dependent Processes

**DOI:** 10.3390/biology12070897

**Published:** 2023-06-22

**Authors:** Victoria Bunik

**Affiliations:** 1Belozersky Institute of Physicochemical Biology, Department of Biokinetics, Lomonosov Moscow State University, 119234 Moscow, Russia; bunik@belozersky.msu.ru; 2Faculty of Bioengineering and Bioinformatics, Lomonosov Moscow State University, 119234 Moscow, Russia; 3Department of Biochemistry, Sechenov University, 119048 Moscow, Russia

**Keywords:** vitamin B1, vitamin B6, vitamin B3, pyridoxal-5′-phosphate, thiamine diphosphate, NAD+, mutation in vitamin-binding protein, Charcot–Marie–Tooth disease, *DHTKD1*, *PDK3*, *PDXK*

## Abstract

**Simple Summary:**

The molecular mechanisms of Charcot–Marie–Tooth (CMT) disease, involving impaired vitamin metabolism and/or actions, are considered in light of the potential therapeutic actions of vitamins B1, B3 and B6 in the disease. Although the different disease cues merge into similar symptoms, identification of disease-specific molecular mechanisms is necessary to develop tailored treatments in personalized medicine.

**Abstract:**

Understanding the molecular mechanisms of neurological disorders is necessary for the development of personalized medicine. When the diagnosis considers not only the disease symptoms, but also their molecular basis, treatments tailored to individual patients may be suggested. Vitamin-responsive neurological disorders are induced by deficiencies in vitamin-dependent processes. These deficiencies may occur due to genetic impairments of proteins whose functions are involved with the vitamins. This review considers the enzymes encoded by the *DHTKD1*, *PDK3* and *PDXK* genes, whose mutations are observed in patients with Charcot–Marie–Tooth (CMT) disease. The enzymes bind or produce the coenzyme forms of vitamins B1 (thiamine diphosphate, ThDP) and B6 (pyridoxal-5′-phosphate, PLP). Alleviation of such disorders through administration of the lacking vitamin or its derivative calls for a better introduction of mechanistic knowledge to medical diagnostics and therapies. Recent data on lower levels of the vitamin B3 derivative, NAD+, in the blood of patients with CMT disease vs. control subjects are also considered in view of the NAD-dependent mechanisms of pathological axonal degeneration, suggesting the therapeutic potential of vitamin B3 in these patients. Thus, improved diagnostics of the underlying causes of CMT disease may allow patients with vitamin-responsive disease forms to benefit from the administration of the vitamins B1, B3, B6, their natural derivatives, or their pharmacological forms.

## 1. Introduction

Pharmacological doses of vitamins may be successfully applied to fight neurological diseases, both acquired and genetically determined [1,2,3,4,5]. The goal of this review is to demonstrate how our understanding of the metabolism and action of vitamins in mammals helps to suggest therapies whose timely application saves genetically, nutritionally and/or aging-compromised organisms from progressive damage. A relatively advanced understanding of mammalian metabolism of vitamin B1 (thiamine), vitamin B3 (nicotinamide, nicotinic acid), and vitamin B6 (pyridoxal, pyridoxine and pyridoxamine), added to the well-known mechanisms of the metabolic action of their coenzyme forms, i.e., thiamine diphosphate (ThDP), NAD+, and pyridoxal-5′-phosphate (PLP), underlies the choice of these vitamins to address the review goal. In view of the wide spectrum of neurological disorders induced by impairments in the metabolism of these vitamins or their coenzyme action, the focus of this review is on vitamin B1-, B3- or B6-related disorders manifesting in Charcot–Marie–Tooth (CMT) disease. When diverse cues converge into the similar pathology, exemplified by axonal degeneration in CMT disease, understanding the different molecular mechanisms of the pathology is essential for finding therapies to counteract the cause of the pathology, not only its symptoms.

Common symptoms due to the compromised metabolic action of each of the vitamins may result from their metabolic interplay. First of all, the coenzyme derivatives of vitamins B1, B3 and B6 function in interconnected metabolic nodes. For instance, de novo production of the major neurotransmitters glutamate and gamma-aminobutyric acid (GABA) from glucose requires the participation of ThDP, PLP, and NAD+ (Figure 1), exposing their critical and interconnected value for neuronal function.

Amphibolic metabolite 2-oxoglutarate is generated from glucose in the mitochondrial tricarboxylic acid cycle. The further ThDP-dependent degradation of 2-oxoglutarate results in energy production. Alternatively, 2-oxoglutarate is required for the biosynthesis of glutamate, which occurs through the PLP-dependent transamination or NAD+-dependent reductive amination of 2-oxoglutarate. Glutamate further generates GABA in the PLP-dependent decarboxylation (Figure 1). This example demonstrates that perturbed levels of any of the involved vitamin derivatives may adversely affect the metabolic network of the major neurotransmitters. Therefore, different regulatory mechanisms exist to support the stable functioning of the metabolic network upon perturbations in specific reactions. In particular, the enzyme pyridoxal kinase, producing the coenzyme form of vitamin B6, is regulated by phosphorylated forms of vitamin B1 [6]. As a result, the interplay of B1, B3, and B6 vitamins in the metabolic pathways essential for neural functions provides a good explanation for the notion that different perturbations in vitamin-dependent processes may converge into axonal degeneration, underlying the neuropathological symptoms of CMT disease. Moreover, exacerbation of the disease symptoms with aging may be contributed by the known age-associated impairments in vitamin status and metabolism [7,8,9,10].

The pathologies induced by perturbed vitamin action or status are often alleviated via administration of the corresponding vitamin, with timely administration preventing irreversible secondary damage [2,11]. Hence, especially at the initial stages of the disease, knowledge on the specific primary impairment is of critical therapeutic significance. This review draws attention to the translational value of basic knowledge of the disease mechanisms by demonstrating the relationship between CMT disease with perturbed vitamin-dependent processes and the therapeutic potential of the vitamins administration. The provided considerations call upon further clinical studies of vitamins B1, B3, and B6 as components of combinatorial therapies for patients with CMT disease associated with insufficiencies of metabolic and/or regulatory reactions for which the vitamins are required.

## 2. General Information on Charcot–Marie–Tooth Disease and Associated Mutations

CMT disease is a group of non-fatal neurological disorders affecting approximately 1 in 2500 people. Despite clinical and genetic heterogeneity, common manifestations of the disease are impairments of the motor and/or sensory peripheral nerves, resulting in muscle weakness and atrophy, accompanied by sensory loss. The neural cells of patients with this disease cannot support the required electrostimulation due to axonal anomalies in the motor and sensory neurons. Impaired peripheral signaling in CMT disease is generally suggested to be due to de-energization of the axon endings, which is linked to mitochondrial abnormality. Dependent on the primary cause, the two major forms of CMT are the demyelinating form and axonal form [12]. Although the convergent symptoms point to interdependence of the pathological events in different forms of CMT disease, identification of its specific molecular mechanisms is crucial for cause-directed therapies. Yet, in autosomal recessive CMT disease, causal genetic variants are identified only in a quarter of all cases.

Approximately 1000 mutations in 90 different genes are known to cause CMT disease, with most of the mutations affecting the cytoskeleton or axonal trafficking [13,14]. Genetic impairments in the mitochondrial proteins represent the most direct pathway to mitochondrial insufficiency associated with CMT disease. The two genes (*DHTKD1*, *PDK3*) encode for the mitochondrial enzymes binding the coenzyme form of vitamin B1 (ThDP), and another gene (*PDXK*) encodes for the producer of the coenzyme form of vitamin B6 (PLP). Mitochondrial metabolism and energy production strongly depend on both the ThDP- and PLP-catalyzed reactions (Figure 1).

In addition to mutated genes, a balance between biosynthesis and degradation of a major mitochondrial redox substrate NAD+ (vitamin B3 derivative) has a significant role in axonal degeneration [15,16]. The link between the NAD+ levels and CMT disease is supported by reduced levels of NAD+ in the blood of CMT patients vs. healthy subjects [17]. Thus, the pathogenic mechanisms of CMT disease, arising due to currently known mutations in the ThDP- or PLP-binding enzymes, are considered below, along with the data from animal and cellular models that demonstrate the molecular mechanisms underlying the potential use of vitamins B1, B3, and B6 as therapeutic agents to overcome CMT pathology.

## 3. CMT Disease upon Mutations of Vitamin-Dependent Enzymes

The known forms of CMT disease due to mutations of genes encoding vitamin-dependent enzymes are summarized in Table 1.

### 3.1. Vitamin-B6-Dependent Pyridoxal Kinase

The *PDXK* gene encodes for pyridoxal kinase (PDXK), catalyzing the ATP-dependent phosphorylation of different vitamers of vitamin B6 (pyridoxal, pyridoxin, pyridoxamine) to their phosphorylated forms. The phosphorylated derivative of pyridoxal, PLP, is a coenzyme of a number of enzymes, including those essential for energy production and neurotransmitter metabolism. The enzymes are exemplified in Figure 1 as transaminases and PLP-dependent decarboxylases.

Two biallelic variants of the *PDXK* gene, shown in Table 1 (c.682G>A and c.659G>A with the protein substitutions Ala228Thr and Arg220Gln, correspondingly), have been identified in five individuals from two unrelated families (OMIM 618511). The variants cause the autosomal recessive CMT disease of 6C type (CMT6C) with primary axonal polyneuropathy and optic atrophy, also called as hereditary motor and sensory neuropathy-type VIC with optic atrophy (HMSN6C) [3]. Distal muscle weakness and atrophy predominantly affecting the lower limbs have an onset in the first decade of life. Optic atrophy and vision loss occur during adulthood.

The substituted amino acids Ala228 and Arg220 are evolutionary conserved residues participating in PDXK interaction with its substrate ATP. Hence, substitutions of these conserved residues impair the substrate binding, resulting in the reduced PDXK activity and low PLP levels observed in the patients. Another homozygous missense mutation in the *PDXK* gene (c.225T>A with the protein substitution Asn75Lys) has been revealed in two children from one family exhibiting symptoms of CMT disease [18]. This substitution decreases the stability of the PDXK dimer and increases protein degradation. As a result, the patient levels of the PDXK activity and PLP are very low.

Oral PLP administration at a dose of 50 mg/kg, known to have none of the neurotoxic effects that are possible with high doses of vitamin B6, increases the levels of PLP in the blood of patients with pathogenic substitutions in PDXK, characterized by impaired production of PLP [3,18]. In CMT disease with the Ala228Thr substitution in PDXK, the PLP replacement scheme has been shown to stably increase PLP levels in patients up to 24 months [3]. Moreover, upon long-term follow-up, an increase in PLP alleviate the symptoms and moderate the progress of CMT disease induced by mutations of *PDXK* gene. The data obtained [3,18] stress the need to screen for potential *PDXK* mutations in patients with autosomal recessive polyneuropathy of early onset, as prompt PLP supplementation may cause long-term improvement in clinical outcomes.

### 3.2. Vitamin-B1-Dependent Enzymes

Vitamin B1 (thiamine), its natural derivative ThDP (cocarboxylase), and pharmacological forms (benfotiamine, sulbutiamine, etc. [19]) are known to exhibit cardio- and neuroprotective actions under a variety of conditions associated with dysfunctional mitochondria [1,20,21,22]. A potential link between thiamine and the CMT disease symptoms is supported by an outbreak of a peripheral neuropathy similar to CMT disease in 88 male prisoners with hyporeflexia/areflexia of the lower extremities, sensory deficit, and motor weakness. The pathology has been diagnosed along with the observation of thiamine deficiency in 80% of the aforementioned prisoners [23].

Regarding the benefits of thiamine and its known pharmacological forms for treating CMT disease, cases of the disease induced by pathogenic mutations in the genes encoding for mitochondrial enzymes regulated by ThDP are of special interest. These are mutations in the *DHTKD1* and *PDK3* genes encoding the mitochondrial enzymes 2-oxoadipate dehydrogenase (OADH) and the neural-tissue-specific isoenzyme 3 of pyruvate dehydrogenase kinase (PDK3), correspondingly (Table 1). ThDP is a coenzyme of OADH and an inhibitor of PDK3 [24,25,26]. Remarkably, ThDP is also a coenzyme of the PDK3 target pyruvate dehydrogenase, which is inactivated by PDK3-catalyzed phosphorylation. As the first component of the key mitochondrial pyruvate dehydrogenase complex (PDC) coupling glycolysis to the mitochondrial metabolism, pyruvate dehydrogenase is indispensable for oxidation of the glucose-derived pyruvate in mitochondria (Figure 1). 

**Table 1 biology-12-00897-t001:** Forms of Charcot–Marie–Tooth disease caused by mutations in the genes encoding vitamin-dependent enzymes.

Gene	Enzyme and Its Vitamin-Related Ligand	Gene Mutation	Protein Variant	Functional Significance of the Mutation	Primarily Affected Pathways	CMT Type	References
*PDXK*	Pyridoxal kinase, producer of the coenzyme form of vitamin B6 (PLP)	c.682G>A c.659G>A c.225T>A	Ala228Thr Arg220Gln Asn75Lys	Perturbed ATP binding, decreased enzyme activity, and low PLP levels are shown. Perturbed ATP binding, decreased enzyme activity, and low PLP levels are shown. Dimer destabilization and increased enzyme degradation are predicted. Very low levels of the enzyme activity and PLP are shown.	Transamination and decarboxylation of amino acids (decreased)	CMT6C (HMSN6C) CMT6C (HMSN6C) CMT6C (HMSN6C), early onset	[3,18]
*PDK3*	Kinase of pyruvate dehydrogenase (isoform 3), inhibited by the coenzyme form of vitamin B1 (ThDP)	c.G473>A	Arg158His	A 5-fold increase in the enzyme activity and a 6-fold stronger binding to PDC are shown. In the *C.elegans* model, perturbed synaptic transmission and decreased ATP production are shown.	Mitochondrial oxidation of the glucose-generated pyruvate (decreased)	CMTX6	[13,27,28]
*DHTKD1*	2-Oxoadipate dehydrogenase, requires the coenzyme form of vitamin B1 (ThDP) for its function	c.1455T>G	No protein from the mutated DNA	Preterm transcription termination at Tyr485 causes a 2-fold decrease in the enzyme mRNA. In HEK293 cells, rapid degradation of the truncated mRNA is shown.	Perturbed degradation of tryptophan and lysine	CMT2Q	[29]

While PDK3-dependent PDC phosphorylation is a well-known regulatory mechanism [30], the role of the OADH-dependent PDC glutarylation is only just emerging [31]. Nevertheless, the finding that OADH is involved in post-translational modifications of PDC in addition to PDK3, is of pathophysiological significance, as excessive protein glutarylation has been observed in a number of pathological states [32].

Molecular mechanisms of mitochondrial dysfunction as a result of CMT disease-associated mutations in the mitochondrial ThDP-regulated OADH and PDK3 (Table 1) are considered below, in view of the therapeutic potential of vitamin B1.

#### 3.2.1. Isoenzyme 3 of Kinase of the ThDP-Dependent Pyruvate Dehydrogenase

Mutation c.G473>A in the *PDK3* gene causes the substitution Arg158His in PDK3, representing the genetic cause of an X-linked dominant form of axonal CMT disease (CMTX6) (OMIM 300905) [13,27,28]. Males are more severely affected by this mutation than females. The symptoms onset occurs within the first 13 years, when the muscle weakness and atrophy, predominantly in lower limbs, and sensory deficits are detected. Some of the affected men have moderate sensorineural hearing loss and perturbed auditory brainstem response.

The substitution Arg158His leads to a hyper-active PDK3 exhibiting an increased affinity to PDC, compared to the wild-type enzyme. As a result, higher levels of phosphorylation and inactivation of cellular PDC are observed in patients carrying this mutation [28]. The metabolic characteristics of fibroblasts and iPSC-derived motor neurons of the patients reveal failures in energy production and mitochondrial function. In vivo models wherein the Arg158His-substituted PDK3 ortholog, known as PDHK2 in *C. elegans*, is knocked-in, or the wild-type and mutated PDK3s are overexpressed in the GABA-ergic motor neurons, recapitulate deficits in mitochondrial function and synaptic neurotransmission, as observed in the cellular studies [13]. Additionally, the models reveal characteristics of CMT disease, such as locomotion defects and signs of progressive neurodegeneration.

ThDP is known to inhibit kinases of pyruvate dehydrogenase [24,25,26]. Hence, patients with CMT disease induced by hyperactive PDK3 may benefit from administration of the PDK3 inhibitor ThDP, which is well-known in medicine and pharmacology as cocarboxylase. Pharmacological lipophilic forms of thiamine (vitamin B1), such as sulbutiamine (enerion) and benfotiamine, are also widely available. Remarkably, ThDP inhibition of PDK3 would increase the activity of pyruvate dehydrogenase by decreasing the enzyme phosphorylation level. Simultaneously, as a coenzyme of pyruvate dehydrogenase, ThDP may activate the PDC-catalyzed oxidation of the glycolytic product pyruvate via an independent mechanism, such as the saturation of the pyruvate dehydrogenase with its coenzyme. As a result, the thiamine administration may counteract the hyperactivity of the Arg158His-substituted PDK3 by inhibiting PDK3 and activating PDC, potentially relieving the CMT disease caused by this mutation. This mechanistic knowledge calls for a clinical evaluation of the therapeutic effect of vitamin B1 or its derivatives in CMT patients with the hyperactive PDK3.

#### 3.2.2. Molecular Mechanisms of CMT Disease Caused by Mutations in the DHTKD1-Encoded ThDP-Dependent 2-Oxoadipate Dehydrogenase

The axonal type of CMT disease type 2Q (OMIM 615025) is described upon heterozygous loss-of-function mutation of the *DHTKD1* gene encoding for the mitochondrial ThDP-dependent OADH [29]. The mutation c.1455T>G in exon 8 of the *DHTKD1* gene causes preterm termination of the transcription at the Tyr485 codon, and a 2-fold decrease in the OADH mRNA. The resulting autosomal dominant form of CMT disease is characterized by muscle atrophy, predominant weakness of the lower limbs, decreased or absent deep tendon reflex, and mild to moderate sensory impairment. The symptom onset is detected from 13 to 25 years.

Mice models of homozygous knockout of the *DHTKD1* gene mimic the CMT disease phenotype, demonstrating the anatomic and functional features of peripheral neuropathy with characteristic perturbations of the motor and sensory functions, axonal degeneration, and muscle atrophy. The phenotype is accompanied by serious metabolic abnormalities, with significant increases in the urine levels of 2-oxoadipate and its transamination product 2-aminoadipate [33].

Pathogenic homozygous *DHTKD1* mutations are associated with strongly increased levels of the OADH substrate 2-oxoadipate and its derivatives (2-aminoadipate, 2-hydroxyadipate), causing severe neurological manifestations, delayed development, chronic diseases of respiratory pathways, and early muscle atrophy [34,35,36]. Nevertheless, in some mutants, clinical manifestations are absent, even when biochemical parameters are perturbed [35,36].

The pathogenicity of heterozygous mutations in the *DHTKD1* gene varies. The mutations may be associated with neuropathies, such as CMT disease and similar states, with autoimmune disease, such as eosinophilic esophagitis, or with epilepsy, but often the same mutations may be present in asymptomatic persons [29,37,38,39]. Most of the nucleotide substitutions in the *DHTKD1* gene are asympthomatic [40]. Thus, understanding molecular basis of pathophysiological impact of the *DHTKD1*-encoded OADH is not straightforward. The impact potentially involves interactions with other pathways, which are considered below.

The current level of characterization of the structure–function relationship in the OADH molecule allows one to predict the pathogenicity of the amino acid substitutions in the enzyme active site and protein–protein interfaces. Structures of recombinant OADH [40,41] and homology modelling of the enzyme complexes with its ligands [42,43] reveal the impact of OADH mutations in the active site and dimeric interface of the enzyme on the enzyme-catalyzed reaction. Besides, the protein residues involved in heterologous interactions upon formation of the OADH multienzyme complex, where OADH catalyzes oxidative decarboxylation of 2-oxoadipate with generation of glutarylCoA and NADH, are also characterized [44,45,46]. This knowledge enables predictions of functional impairments due to mutations affecting the complex formation. Nevertheless, several considerations are important to note, regarding the varied pathophysiological impact of the same DHTKD1 mutations in different carriers of a mutation. First, the catalytic function of OADH is redundant, and could well be substituted by an ubiquitously expressed OADH isoenzyme, 2-oxoglutarate dehydrogenase (OGDH). In vitro, OGDH catalyzes oxidative decarboxylation of 2-oxoadipate at a rate of up to 50% of that of oxidative decarboxylation of 2-oxoglutarate, with OGDH expression usually exceeding that of OADH [47]. Second, in animal tissues, OADH undergoes post-translational modifications strongly affecting the enzyme structure, compared to that of the characterized recombinant enzyme, which may correspond to the existence of an as yet unidentified enzyme function [48]. Third, the biochemical effects of OADH mutations, i.e., increases in the 2-oxo/aminoadipate levels, are not necessarily accompanied by clinical symptoms [35]. These considerations suggest that the (patho)physiological impact of OADH relies on the OADH-specific interplay with other proteins/pathways, rather than solely on OADH catalytic function.

This assumption is supported by the data on the most characterized associations of OADH with insulin resistance, obesity and type II diabetes. The data not only reveal the significance of genetic background in the biological impact of OADH, but also suggest that perturbed glucose metabolism is a molecular mechanism underlying OADH-associated neurological disorders. Indeed, increased 2-oxoadipate and/or 2-aminoadipate levels in the mice knockouts of *DHTKD1* [33,49,50] cause an insulin resistance phenotype [33]. Liver expression of *DHTKD1* is the major regulator of the level of 2-aminoadipate, the transamination sibling of the OADH substrate 2-oxoglutarate, in serum [51]. Dependent on external (diet) and genetic (different alleles) factors, *DHTKD1* gene expression also correlates with levels of glucose and cholesterol, and the diabetic status [51]. In the blood plasma of humans and mice, glucose levels are negatively correlated with those of 2-aminoadipate, while the levels of 2-aminoadipate and insulin correlate positively [52]. When β-cells of the pancreas are treated with 2-aminoadipate, more insulin is secreted, corresponding with increased insulin secretion upon the accumulation of 2-aminoadipate in the pancreas, as observed in mice on a high-fat diet [52]. At the same time, mice knockouts of the *Csf2* gene (controlling the function of granulocytes and macrophages) are obese without diabetic symptoms. Remarkably, these mice exhibit increased expression of *DHTKD1* and decreased 2-aminoadipate levels compared to wild-type mice [53]. *DHTKD1* expression is also increased by a natural compound mangiferin, preventing obesity [54]. Vitamin B1 (thiamine), which is a precursor of the OADH coenzyme ThDP, upregulates OADH activity in the rat cerebellum [55].

Thus, relatively low expression of *DHTKD1* may be an early indicator of obesity and type II diabetes, while increased expression of *DHTKD1* is associated with clinical improvement of such states [56]. Pharmacological regulation (e.g., mangiferin, vitamin B1) or genetic factors (exemplified by knockout of the *Csf2* gene) may increase *DHTKD* expression. Compensating for functional impairments in the OADH mutants, this conditional increase in *DHTKD* expression may contribute to the heterogeneity of the mutant *DHTKD1* phenotypes. The available data thus imply that pharmacological upregulation of OADH expression and/or activity via administration of vitamin B1 may be of therapeutic value.

If phenotypic manifestations of the *DHTKD1* mutations, particularly in CMT disease, depend on the function of other genes, a question arises what these other genes are. To answer this question, a closer look at the metabolic pathways potentially affected by OADH function is required (Figure 2).

Participating in the production of glutaryl-CoA for protein glutarylation, OADH is involved in metabolic regulation via post-translational modifications through the acylation of protein lysine residues. The recent discovery of the glutarylation of the pyruvate dehydrogenase complex, a key player in the oxidative glucose metabolism, provides a mechanistic link between the OADH function and oxidative glucose metabolism [31], as characterized in the association studies discussed above. Depending on the pathophysiological state, short-term inhibition of OADH in rat brain may cause a long-term increase in enzyme expression, which is associated with increased glutarylation of the brain proteins and elevated expression of deglutarylase sirtuin 5 [31].

Furthermore, the existence of an N-terminus-truncated isoform of OADH in mammalian tissues, lacking the protein part involved in the formation of a multienzyme complex but comprising the active site [48], agrees with our earlier suggestion [44] that OADH may also have some functions outside of the multienzyme complex. An example of such functions is glyoxylate detoxication, which involves non-oxidative decarboxylation of 2-oxoacids via their ThDP-dependent dehydrogenases.

Finally, functioning in the tryptophan degradation pathway, OADH is linked to the de novo biosynthesis of NAD+ and its signaling derivatives from tryptophan (Figure 2). Their interplay is supported by perturbed nicotinamide (vitamin B3) homeostasis upon OADH inhibition [57]. Perturbed NAD+ metabolism has also been shown in independent animal and cellular models of OADH impairment [33,58], further exposing the role of NAD+ in CMT disease arising from the *DHTKD1* mutations. In fact, mutations and knockouts of the *DHTKD1* gene decrease the NAD pool and energy metabolism [33,58]. Combined with the data on the perturbed pool of vitamin B3 in the OADH-inhibited cells [57] and neurological manifestations of impaired tryptophan catabolism [59], the available data argue for the significance of NAD+-dependent pathways in the pathophysiological contribution of OADH function. Hence, in addition to the OADH activator thiamine/ThDP, administration of vitamin B3, especially in the form of its derivative nicotinamide riboside, which helps in overcoming NAD+ biosynthetic hurdles [60,61], may be promising for alleviating the neurological symptoms of CMT disease caused by the *DHTKD1* mutations. The underlying molecular mechanisms are considered in more detail in Section 4.

### 3.3. The Shared Genetic Origin of the Vitamin-Related Forms of CMT and Other Neurological Disorders

The above-considered genes involved in CMT disease (Table 1) are also known to be associated with other neurological disorders. Fourteen rare heterozygous mutations of the *DHTKD1* gene have been observed in amyotrophic lateral sclerosis (ALS), a fatal neurodegenerative disorder characterized by progressive loss of the upper and lower motor neurons [62]. These results suggest that rare heterozygous *DHTKD1* variants may contribute to the ALS phenotype, and possibly to pathogenesis.

The homozygous mutation of the *DHTKD1* gene has also been found in a patient with infantile onset of spinal muscular atrophy, followed by a psychomotor delay at an age of 5 years. In this case, the pathogenic variant of OADH with the substitution Gly729Arg causes the accumulation of 2-oxoadipate (OMIM 204750) [63]. Remarkably, spinal muscular atrophy due to the accumulation of 2-oxoadipate is also observed upon the mutation of the mitochondrial dicarboxylate transporter of 2-oxoadipate, SLC25A21 [64]. The different molecular mechanisms underlying the 2-oxoadipate accumulation predict different responses of the aciduria to the administration of vitamin B1. Activation of OADH [64] by ThDP would not help to transport 2-oxoadipate into the mitochondria with the mutated SLC25A21. However, if 2-oxoadipate accumulation is caused by the functionally impaired variant of OADH, its coenzyme ThDP may improve the enzyme functioning. As shown in studies of other ThDP-dependent 2-oxoacid dehydrogenases, this improvement may occur not only through increased saturation of the enzyme with the coenzyme, but also through the ThDP-dependent stabilization of catalytically competent protein conformations [65,66]. Thus, knowledge on the specific molecular mechanism of 2-oxoadipic aciduria is necessary for developing appropriate therapies.

A variant of *PDXK* gene has been reported to be associated with Parkinson’s disease in European cohorts; this discovery came after whole-genome expression profiling of substantia nigra neurons isolated from Parkinson’s disease patients [67,68]. Short reports from replica studies on other cohorts [69,70] do not confirm this result. However, the experimental design, especially regarding the samples’ preparation, is not sufficiently specified in these short reports. The discrepancy in results is likely due to the different resolution of analytical procedures and/or cell-specific effects, themselves dependent on the tissue sampling. In a later work, inhibition of PDXK in a *Drosophila* model recapitulated symptoms of Parkinson’s disease, such as decreased lifespan and loss of locomotor function [71]. Moreover, in a rat model of spinal cord injury, *PDXK* overexpression in the motor cortex improved neuronal growth and survival, as well as the locomotor function in the hindlimb. In the same study, knockout of the microRNA (MiR339) that blocks PDXK translation was shown to decrease apoptosis of cortical neurons, thereby increasing neurite outgrowth [72].

A genome-wide meta-analysis has associated a variant of the *PDK3* gene with insomnia [73]. The details are available from the GWAS catalog database, at https://www.ebi.ac.uk/gwas/variants/rs1207613, accessed on 21 June 2023. It is interesting in this regard that administration of thiamine, whose diphosphate derivative ThDP inhibits PDK3, regulates phosphorylation of pyruvate dehydrogenase and acetylation of glutamate dehydrogenase in the rat brain in a day-time-dependent manner [74,75]. These data provide molecular mechanisms for the known associations of sleeping patterns with thiamine and its pharmacological forms [76,77].

## 4. CMT Disease, Aging and the Vitamin B3 Derivative NAD+

The symptoms of CMT disease may appear at an earlier or later age. Accordingly, the disease progress varies significantly. Many patients remain active and have a normal life expectancy, but severe cases have also been known, in which respiratory difficulties may be lethal [78]. Increased manifestations of CMT disease with age coincide with the age-dependent decrease in cellular NAD+ levels [79,80,81,82,83]. The link deserves more attention, especially regarding CMT disease developed in people with a mutated *DHTKD1* gene, encoding the OADH protein that is linked to the metabolic and signaling pathways involving NAD+ (Figure 2).

Decreasing NAD+ levels in cells and tissues upon normal aging are considered to be due to the aging-induced drop in NAD+ biosynthesis and/or the increase in NAD+ degradation. Apart from its well-known function of being a substrate of redox reactions, NAD+ is involved in the post-translational regulation of proteins and DNA damage response, as a substrate of sirtuins and poly-(ADP-ribose) polymerase (PARP), respectively [84]. NAD+ is also used in the biosynthesis of signaling molecules by nucleosidases CD38, CD157, and SARM1, regulating calcium homeostasis and NAD+ metabolism [81,82]. In particular, production of adenosine diphosphate ribose and its cyclic form in the SARM1-catalyzed reaction of NAD+ degradation (NADase reaction) is a sign of axonal degeneration. The NADase activity of SARM1 is greatly stimulated by axonal damage. The loss of intracellular NAD+ upon such damage may induce SARM1 activation, as high concentrations of NAD+ inhibit SARM1 [85]. When the loss of NAD+ due to axonal damage activates SARM1, axonal self-destruction is initiated, which is associated with energy failure, Ca^2+^ overload, and the activation of proteolysis. In contrast, increasing NAD+ levels may not only attenuate the activation of SARM1 upon neurodegenerative insults, but also improve energy status, supporting calcium homeostasis and associated events [85]. The role of increased NAD+ biosynthesis in the pathological self-destruction of damaged axons, known as Wallerian degeneration, has been demonstrated by studies of the mouse mutation in the *WLD* gene. When the gene is mutated to *WLD^S^* (Wallerian degeneration slow), the pathological axon degeneration under a number of neurodegenerative conditions, including CMT disease, is suppressed. Remarkably, the *WLD* gene is mapped to the distal region of the mouse chromosome 4, with an axonal form of CMT disease mapped to the homologous location in humans, 1p36. This region includes the gene encoding an enzyme for NAD+ biosynthesis, nicotinamide/nicotinic acid mononucleotide adenylyltransferase 1 (NMNAT, Figure 2), whose mutation causes the axonal form of CMT disease (OMIM 608700). In the *WLD^S^* mice, axonal NAD+ biosynthesis is activated, as the *WLD^S^* mutation enables expression of a chimeric protein (WLD^S^), comprising NMNAT1 [86]. In the same work, *SIRT1*-encoded sirtuin 1 is shown to be a downstream activator of the increased activity of NMNAT1. Thus, fighting NAD+ depletion in axonal degeneration is suggested as a therapeutic approach to CMT disease [87].

In this regard, the observation of decreased NAD+ levels in the blood of patients with CMT disease, compared to healthy subjects [17], is remarkable. This NAD+ decrease supports independent studies on the role of NAD+ in pathological axon degeneration, which is observed in CMT disease [16,85]. The positive action of the administration of different NAD+ precursors is known in a number of pathologies with neurological manifestations [81,82]. In particular, one NAD+ precursor, nicotinamide riboside, has been shown to counteract excitotoxicity-induced neurodegeneration [15]. Thus, the available data suggest vitamin B3 and its derivatives to be of therapeutic potential in CMT disease.

## 5. The Tissue-Specific Impact of Gene Mutations Perturbing Vitamin-Dependent Processes

This review deals with the perturbed processes dependent on vitamins B1, B3, and B6, which result in non-fatal peripheral polyneuropathy such as CMT disease. The considered gene mutations cause peripheral perturbations affecting axonal function and morphology. Remarkably, when the status and metabolism of the same vitamins is perturbed in the brain, much more severe neurological disorders develop. For example, mutations in the *PNPO*- or *PROSC*-encoded proteins of vitamin B6 metabolism cause severe epilepsies in newborns [88]. Moreover, epileptic seizures are also known in animal models with dysregulated expression of *PDXK* in the brain [89]. However, with the *PDXK* mutations leading to CMT disease, B6-dependent impairments are limited to peripheral signaling. It may therefore be suggested that this peripheral pathological manifestation is due to the axon-specific features of metabolism and the heterological interaction of the mutated protein. It has also been observed in independent studies that the regulation of *PDXK* expression differs in the peripheral tissues compared to the brain [3].

Similarly, a well-known consequence of the perturbed functioning of the ThDP-dependent enzymes in the brain is Wernicke’s encephalopathy. Its unusual presentation may include epileptic seizures, as observed in pregnancy [90,91]. An outbreak of the life-threatening Wernicke’s encephalopathy has occurred in infants consuming a thiamine-deficient soy-based formula [11]. In the long-term follow-up of the eight initial survivors, six have developed severe epilepsy [92]. These cases suggest that untreated deficiency of vitamin B1 in the brain may progress from Wernicke’s encephalopathy to epilepsy, probably interacting with such factors as increased nutritional and metabolic demand in pregnant or developing organisms. Severe epileptic seizures have also been reported in some patients with the *DHTKD1* mutations, although interaction with non-identified co-morbidities cannot be excluded in these cases [36].

A characteristic feature of seizures arising from impaired metabolism of vitamins B1 or B6 in the brain is a lack of response to conventional anticonvulsant treatments. Instead, the seizures disappear upon administration of the corresponding vitamins, in a manner similar to the less severe symptoms of CMT disease caused by axonal impairments of vitamin-dependent processes.

## 6. Therapeutic Doses and Pharmacological Forms of Vitamins B1, B3, and B6

Current therapies to treat CMT disease are mostly symptomatic. However, nutritional approaches (including curcumin, likely alleviating the unfolded protein response, and vitamin C, which has been progressed to a phase II clinical trial) have also been considered [93]. This review indicates that knowledge on the molecular mechanisms of CMT disease, partially shared with other neurodegenerative diseases, may be used to develop therapies utilizing administration of pharmacological doses and/or forms of vitamins B1, B3, and B6 in some CMT patients. Importantly, administration of these vitamins or their derivatives is used in medicine to treat various disorders other than CMT disease. From these other applications, information on their safety and tolerance has been gained. Multiple membrane-penetrating forms of vitamin B1 exist to obviate transporter-imposed limitations on the intracellular vitamin content [19]. A recent single-site blinded phase IIa randomized placebo-controlled pilot trial of benfotiamine in early-stage Alzheimer’s disease patients has produced promising results. The study shows that benfotiamine at a dose of 600 mg per day is safe and very well tolerated in patients with early Alzheimer’s disease [1]. Even higher doses of thiamine or its tetrahydrofurfuryl disulfide (fursultiamine), i.e., up to several g per day, have been safely applied in animals and humans (as reviewed in [75]). An analysis of the different schemes of thiamine administration to counteract genetic defects of thiamine metabolism refers to a maximum dose of 1.5 g per day, with therapeutic effects usually observed at 10–40 mg/kg per day [94].

Unlike vitamin B1, whose oral administration is usually very well tolerated even at the high doses mentioned above, high doses of vitamin B6 are toxic. Owing to this, vitamin B6 must be taken with much more caution. Usually, administration of pyridoxin vitamer is preferred, as pyridoxal and PLP possess a chemically reactive aldehyde group that modifies proteins and metabolites in vivo. However, in some cases, PLP exhibits a better effect than pyridoxine when the two are compared at a daily dose of 30 mg/kg [95]. Treatments of pyridoxin-responsive epilepsies due to genetic defects of the vitamin B6 metabolism employ doses of 5–15 mg/kg/day. The upper daily limit is 200–300 mg; it is recommended that this amount be introduced in the two doses [96].

Regarding the need to increase the levels of NAD+ that are decreased in CMT patients [17], current trends in stimulating NAD biosynthesis involve administration of the NAD+ biosynthetic precursors nicotinamide riboside and nicotinamide mononucleotide, instead of the traditional precursor nicotinamide. The upper intake level for nicotinamide mononucleotide is estimated at 900 mg daily for a person weighing 60 kg; oral administration of nicotinamide riboside in healthy men and women has been studied at doses of up to 1 g per day [97].

## 7. Conclusions

CMT disease may develop due to a great number of gene mutations, some of which affect vitamin-dependent processes. Ensuing perturbations in the mutation-affected actions of vitamins B1 and B6 may be counteracted using administration of the lacking vitamin or its derivative.

Vitamin B3 is a precursor of NAD+. In aging, the increased NAD+ degradation in the DNA damage response and sirtuin-dependent pathways is not properly compensated for by NAD+ biosynthesis. This imbalance may contribute to increased manifestations of CMT disease with age, which are linked to the NAD-dependent regulation of axonal degeneration. The decreased NAD+ levels in the blood of CMT patients, compared to healthy subjects, suggest that administration of NAD+ biosynthetic intermediates, such as nicotinamide riboside or nicotinamide mononucleotide, may decelerate the progression of CMT disease with age.

## Figures and Tables

**Figure 1 biology-12-00897-f001:**
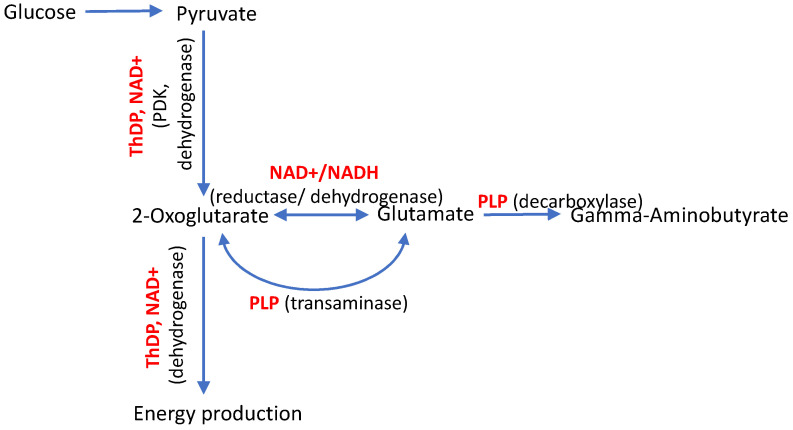
Interplay of natural derivatives of vitamin B1, B3, B6, i.e., thiamine diphosphate (ThDP), oxidized nicotinamide adenine dinucleotide (NAD+), and pyridoxal-5′-phosphate (PLP), correspondingly, in the de novo production of major neurotransmitters glutamate and gamma-aminobutyrate from glucose. The vitamin-dependent enzymes are mentioned in parentheses alongside the type of the catalyzed reactions. PDK stands for pyruvate dehydrogenase kinase, the ThDP-regulated enzyme inactivating the ThDP-dependent pyruvate dehydrogenase through phosphorylation.

**Figure 2 biology-12-00897-f002:**
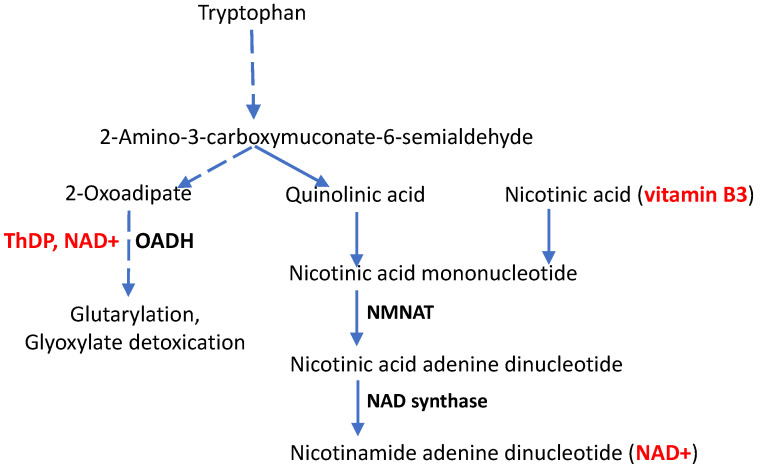
Participation of *DHTKD1*-encoded 2-oxoadipate dehydrogenase (OADH) in the tryptophan catabolism pathway, as an alternative to de novo NAD+ biosynthesis from tryptophan. Dashed lines represent multistep processes, the vitamins and their natural derivatives are shown in bold red, and the enzymes mentioned in the text are shown in bold black text. Nicotinamide/nicotinic acid mononucleotide adenylyl transferase (NMNAT) and NAD synthase catalyze the indicated steps of NAD+ biosynthesis.

## Data Availability

Not applicable.

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
