# Peer review of "The Therapeutic Potential of Vitamins B1, B3 and B6 in Charcot–Marie–Tooth Disease with the Compromised Status of Vitamin-Dependent Processes"

_biology, 2023, doi:10.3390/biology12070897_

Round 1

Reviewer 1 Report

The authors have carried out an exhaustive and deep study both to explain the molecular nature of CMT disease and the role that taking vitamins B1 B3 and B6 can have to alleviate the symptoms of this disease, delving into the mechanisms of action of each vitamin.

It is a very complete report.

Author Response

Thank you for your opinion.

Reviewer 2 Report

This review manuscript is fine and summarizes key aspects about vitamin-related metabolism and their potential use as therapeutic purpose for patients with CMT and specific genetic subtypes. The use of cofactors and some B group-related vitamins may represent a low cost and highly accessible therapy during the follow-up of patients with CMT. Some points should be evaluated by the authors: 

1. The author use the abbreviation in their introduction correctly, but in other parts of the manuscript the author describes the words or terms another time (e.g., PDXK, PLP). 

2. An important aspect in the manuscript is that the author should emphasize that the idea regarding the therapeutic use of vitamins and cofactors in the treatment of inherited neuropathies is based mainly in pathophysiological-related mechanisms and limited in most cases to small case series or only animal models. There are no large double-blinded randomized placebo-based trials. 

3. In the paragraph associated with line 228, I recommend authors to add and briefly discuss the new motor neuron disease phenotype recently described due to DHTKD1 variants associated with Amyotrophic Lateral Sclerosis (Genes (Basel) 2021;13(1):84).

No specific comments regarding English language quality or skills. 

Author Response

Thank you very much for your criticism, which I address as follows.

  1. Corrected where found. I would like, however, to specify that there are mentioning of genes (in italic font) and enzymes (standard font). Because of this, the first mentioning of, e.g., PDXK gene may be followed by deciphering of PDXK enzyme as pyridoxal kinase. Besides, according to some requirements, in the figure legends the abbreviations must be deciphered separately, as the figures are supposed to be self-sufficient for examination.
  2. In the simple summary and in the introduction, I wrote about the therapeutic potential as a "translational deduction": "Molecular mechanisms of Charcot-Marie-Tooth (CMT) disease involving impaired vitamin metabolism and/or action, are considered in view of potential therapeutic action of vitamins B1, B3 and B6 in the disease." "This review draws attention to the translational value of the basic knowledge on the disease mechanisms by demonstrating the relationship between CMT disease, perturbed vitamin-dependent processes and the therapeutic potential of the vitamins administration." In the revision, I extended the Introduction notion by adding: "The provided considerations call upon further clinical studies of vitamins B1, B3, B6 as efficient components of combinatorial therapies for patients with CMT disease associated with insufficiencies of the metabolic and/or regulatory reactions for which the vitamins are required."
  3. Thank you for the remark on the DHTKD1 in ALS. In view ofcomments of  other reviewers, I extended this as an additional subsection of section 3.

Reviewer 3 Report

The authors describe the role of vitamin B1, B3 and B6 in a subgroup of CMTs. It is a good overview over the biochemistry and pathophysiology of vitamin deficiency in humans.

The title is quite misleading suggesting that the article deals with multiple vitamins and the whole entity of CMT. Instead the author describes a small subgroup of CMT patients in whom these vitamins are involved. The title should be more precise.

The paper precisely describes the pathophysiology of a lack of these vitamins. The clinical effect of supplementation and the existance or non-existance of clinical study supporting this should be described more in detail . 

It would be useful to describe in detail the effect of vitamin deficiency as well as the effect of an overdosage in acquired as well as in hereditary forms. The impression the a mixture of vitamins is helpful irrespective of the precise cause of the disease should be avoided. 

Author Response

I thank the reviewer for the careful reading of the manuscript and critical comments.

In the revised version, the title is specified regarding the vitamins ("Therapeutic potential of vitamins B1, B3 and B6..."). Regarding the CMT disease, the title states "Charcot-Marie-Tooth disease with the compromised status of the vitamin-dependent processes".

In the simple summary and in the introduction, I wrote about the therapeutic potential as a "translational deduction":

"Molecular mechanisms of Charcot-Marie-Tooth (CMT) disease involving impaired vitamin metabolism and/or action, are considered in view of potential therapeutic action of vitamins B1, B3 and B6 in the disease."

"This review draws attention to the translational value of the basic knowledge on the disease mechanisms by demonstrating the relationship between CMT disease, perturbed vitamin-dependent processes and the therapeutic potential of the vitamins administration."

In the revision, I extended the Introduction notion by adding: "The provided considerations call upon further clinical studies of vitamins B1, B3, B6 as efficient components of combinatorial therapies for patients with CMT disease associated with insufficiencies of the metabolic and/or regulatory reactions for which the vitamins are required."

The review  goal does not imply to describe in details the consequences of the  deficiencies in vitamins B1, B3, B6, as there are significant differences of these consequences, dependent on exact molecular mechanisms. In so far, mechanisms and consequences of deficient metabolism and action of each vitamin would require a separate review. According to my current review goal, I consider the mechanisms relevant for the CMT disease with impaired status of the B1 and B6-dependent enzymes, as well as the published fact that B3 derivative NAD is decreased in CMT patients.

The section on therapeutic dosages of the considered vitamins/precursors is introduced into the revised version.

Reviewer 4 Report

My suggestions:

1. When authors mention the genetic factors of CMT, they may briefly mention the genetic overlap between CMT and other diseases, such as ALS. 

2. In Chapter 3 I would add a table on the mentioned genes, their function, significant mutations, and their impact on CMT.

3. A figure on the mentioned gene (domain structure) with the location of the significant mutations would be useful. 

4. I would add a pathway figure on PDK3 dysfunctions, and how they could impact CMT-related pathways.

5. I would add a short chapter on possible therapeutic approaches to vitamin deficiency-related CMT.

Author Response

Many thanks to the reviewer for the helpful criticism. I have addressed it in the revised version as follows.

  1. Additional section of Chapter 3 is provided, mentioning the involvement of considered genes  in other neurological disorders
  2. The requested  table is added
  3. The figure with the gene structure is not provided, as this figure would not add much to the information which is already given in the inserted table. The gene kjapping would be useful, if CMT were associated with many mutations of each gene. This is not the case. 
  4. We introduced PDK into Figure 1
  5. The chapter with applied dosages of the vitamins is added

Round 2

Reviewer 4 Report

Modifications seem great. Thank you.